# Assessing emergency healthcare accessibility in the Salton Sea region of Imperial County, California

**Preeti Juturu** [iD] *

School of Public Policy and Department of Economics, University of California, Riverside, California, United States of America

* pjutu001@ucr.edu

## Abstract

The area surrounding California's Salton Sea, which lies within Riverside and Imperial counties, has particularly negative health outcomes. Imperial County, a primarily rural region that encompasses the lake, has pediatric asthma-related emergency healthcare visits that double the state average. This paper seeks to assess the level of emergency healthcare access in the Salton Sea region of Imperial County, drawing from spatial science methods. For this study, the "Salton Sea region" is defined as all Imperial County census tracts that include the Salton Sea within its boundaries. To measure "access," this study calculated driving travel times from census tracts to hospitals within Imperial County rather than Euclidean distance to account for geography and urban infrastructures such as road networks and traffic conditions. This study also used the Rational Agent Access Model, or RAAM, to assess access. RAAM scores account for the supply and demand for hospitals in addition to travel times. Results showed that the average travel time for Salton Sea region residents to drive to Imperial County emergency healthcare facilities ranged from 50–61 minutes, compared to 14–20 minutes for other Imperial County tracts. RAAM scores, compared to other Imperial County tracts, were about 30% higher in the Salton Sea region, meaning that healthcare supply is limited in the region. State and county policy should account for spatial inaccessibility to healthcare institutions in order to address emergency healthcare access.

**Data Availability Statement:** The data underlying the results presented in the study is openly available in Zendo at 10.5281/zenodo.4569261, reference number 4569261.

## Introduction

California's Salton Sea, located in the Imperial and Coachella valleys, is the largest lake in California. As a result of agricultural runoff, pollutants and increasing temperatures, the lake's ecosystem has deteriorated drastically over time and the lake continues to reduce in size, increasing the likelihood of dust pollution in the area. Atmospheric particulate matter in the Salton Sea region exceeds California and National Ambient Air Quality Standards [1], which poses environmental health concerns for residents. Exposure to particulate matter can worsen lung disease and exacerbate pre-existing lung issues [2] can lead to high rates emergency healthcare visits.

**Funding:** The author received no specific funding for this work.

**Competing interests:** The author has declared that no competing interests exist.

Imperial County, one of the counties that encompasses the lake, is disproportionately impacted by both dust pollution and atmospheric particulate matter. Imperial County is a primarily rural region in California and exceeds state standards for particulate matter in the air, consistently having 10 micrometers, or PM10, of particulate matter in the air [3]. PM10 poses severe negative health outcomes such as decreased lung function and asthma attacks [4]. High levels of particulate matter in the air, as well as air pollution from agricultural burning, automobile exhaust and factory emissions from surrounding regions, have contributed to high asthma rates in Imperial County [5]. Approximately 15.1% of Imperial County's population is affected by asthma [6], with pediatric asthma-related emergency room visits and hospitalizations that double the California state average [7].

Additionally, Imperial County residents lack resources to mitigate and manage asthma-related symptoms and complications. Imperial County, driven by its agricultural production, is particularly impoverished, with 21.4% of the population living below the poverty line. The California state average for the population living below the poverty line is 12.8%, making Imperial County well above average [8]. Residents in the region surrounding the Salton Sea face additional disparities, as Census tracts within close proximity to the Salton Sea within Imperial County have average median household incomes that range from $28,000 to $31,593 [8]. This poses an additional challenge for individuals hoping to receive preventative treatment for asthma-related conditions, which may exacerbate the need for emergency care services.

Policymakers have attempted to address the health disparities and environmental health issues in the region in a variety of ways. There have been legislative efforts to obtain funding for air quality improvement initiatives in the region [9], and the establishment of the Imperial County Community Air Monitoring Project has helped to address air pollution by monitoring air quality and increasing accessibility to air quality data for residents to use [5]. By tracking air pollution, the California Department of Public Health is able to gauge environmental health of the area, identify "hot spots" and create policies to avoid areas with high levels of pollution. There have also been efforts to maintain the Salton Sea's ecosystem and ensure that it does not continue to reduce in size, such as the Salton Sea Management Program (SSMP) established by former California Governor Jerry Brown. Projects that came out of this program include dust-suppression initiatives as well as habitat restoration efforts to preserve native wildlife [10]. The Salton Sea Authority, another entity that emerged to address Salton Sea restoration efforts, was formed by Assembly Bill 71 and established the Salton Sea Financial Feasibility Action Plan (FFAP), which allocated funding toward Salton Sea restoration projects [11].

Overall, most policies that address the environmental and health concerns in the region are specific to habitat restoration and dust suppression efforts. This is a common theme in academic literature as well, with studies focused on assessing environmental hazards in the Salton Sea and surrounding vicinity. Though these efforts are significant in resolving regional environmental health risks in the long run, they fail to address the emergency healthcare needs of Imperial County residents. This study attempts to fill gaps in literature surrounding health in Imperial County and the Salton Sea region and bring emergency healthcare access to the forefront of regional policymaking. Geographic unavailability of healthcare services is a prominent issue in rural communities, with rural counties having fewer healthcare facilities and higher travel times than urban areas [12]. Academic literature on access to emergency healthcare services in rural regions is limited in nature, as most work focuses on the number of emergency healthcare services and emergency health service use patterns. These works have noted that rural regions tend to have fewer emergency healthcare services, and emergency health service usage is higher in rural regions than urban regions [13].

Though highlighting the lack of access to emergency healthcare services in rural areas is important, it is important to empirically measure the fundamental level of access individuals

have to existing emergency healthcare services, which is spatial access. This paper seeks to assess the level of emergency healthcare access within Imperial County, with a particular emphasis on the Salton Sea region of the county. This is due to the increased need individuals residing in the region may have due to higher exposure to environmental health hazards. Therefore, this work is unique in that it seeks to determine if the region with a higher need for emergency healthcare services, based on environmental health risk, has higher or lower access compared to surrounding rural regions. This work also draws from spatial data science and geospatial methods, as it utilizes the Rational Agent Access Model (RAAM) to assess spatial access to emergency healthcare facilities in Imperial County. This model assesses spatial access to emergency healthcare facilities by considering both traditional spatial access measures (e.g., travel times) and economic behavioral theory. This spatial accessibility model was developed in 2019 and has been utilized in only one academic paper [14]. By using the RAAM model, this work is able to quantify social phenomenon and behavioral choices that affect a rational agent's decision in seeking particular resources over others. By assessing emergency healthcare access using newly developed tooling, this work furthers the body of knowledge surrounding this model and may be used to identify its benefits and limitations.

## Materials and methods

There are two main objectives of this research. The first objective is to assess accessibility to emergency healthcare facilities in Imperial County using different spatial measures of "access," with a particular focus on the Salton Sea region. This study utilized spatial analysis to assess "access" to emergency healthcare in Imperial County and used spatial analytical tools as well as a spatial accessibility model to determine "access." Dimensions of access can be aspatial or spatial; spatial access is typically measured using travel times and focuses on geographic barriers between supply and demand, whereas aspatial access measures tend to assess demographic factors [15]. Measuring spatial access requires the usage of geographic data and various geospatial models that use travel times and population density to compute accurate measures of accessibility. Spatial access is the most fundamental determinant of accessing resources, as individuals must be able to physically reach a service in an efficient manner. Once physically at the service in question, aspatial barriers impact individuals at a higher degree; aspatial measures in the case of a hospital and healthcare include linguistic barriers, socioeconomic status, insurance status and racial demographics which may influence the way medical professionals treat an individual.

### Computing travel times

Assessing spatial access to services such as emergency healthcare facilities require geographical information and location data. This study used the 2010 Imperial County shapefile from the U.S. Census Bureau as a foundation for spatial analysis. This study also required the geographical locations and information of emergency healthcare facilities in the county to compute spatial access from each census tract in the county to each facility. According to the California Department of Health Care Services, Imperial County has only two hospitals that have emergency healthcare capacities: El Centro Regional Medical Center in El Centro, and Pioneers Memorial Hospital in Brawley [16].

**Travel times versus Euclidean distance.** The second objective of this paper is to use travel times and the Rational Agent Access Model (RAAM) to assess access as opposed to Euclidean distance. Euclidean distance can be viewed as "straight line distance," where the physical distance is calculated between one location to another without regard to geographic factors, road networks or altitudes. Euclidean distance is a common measure of spatial accessibility in many

research studies, as it is simple to compute and less costly. Travel times are a more complex and costly method of assessing access to services, as computation requires the acquisition of road network data, traffic patterns and urban-spatial configuration [14]. In the case of spatial healthcare accessibility, studies have shown that if used to determine non-emergency travel, Euclidean distance is inconsequential when compared to travel times [17].

This work, however, seeks to determine the level of emergency healthcare access in the county as opposed to just healthcare facility access. Though Euclidean distance is a helpful measure to understand general access to facilities, it does not account for realistic barriers to emergency healthcare facilities such as road conditions and networks which determine the timeliness of receiving healthcare. Therefore, this study deemed travel times as the most appropriate method of assessing spatial access to emergency healthcare facilities. In order to generate and analyze travel times, this study sought to assess travel times from the centermost point of each census tract to each hospital in the county, known as a centroid. This study also differentiated between census tracts in the county, with the Salton Sea region being defined as all Imperial County census tracts that include the Salton Sea within its boundaries. This would include travel times for census tracts with the following GEOIDs: 1500000US060250123021, 1500000US060250123022, 1500000US060250124001, 1500000US060250123023, 1500000US060250123011, 1500000US060250124002, 1500000US060250101021.

This study generated census tract centroids and travel times from each census tract in Imperial County to each emergency healthcare facility in the county. Travel times are measured in minutes throughout this study. Since travel times are determined by road networks as well as by the time of day an individual is traveling to a location, this study used a reference time of 3:00 PM on a Monday as the starting time, as approximately 3:00 to 4:00 PM typically has a moderate amount of traffic congestion [18]. This reference time is used as a case study, as this study used ArcGIS Online to compute travel times as it was most efficient to calculate values in this manner. Times can also be generated using the Python Spatial Analysis Package (PySAL), an open-source spatial data science library that generates identical datasets [19]. If the travel times, using moderate traffic times, were high, hospital access would potentially be worse during peak traffic hours. This study also used rural travel times as opposed to normal travel times to assess travel times. Rural travel times model the movement of vehicles in a way that, unlike normal travel times, does not remove the possibility of driving on roads that are unpaved.

It is important to note that though the reference time is a static time frame, which may arguably be limiting in nature considering the unpredictability of emergency health situations, rural regions such as Imperial County experience constant traffic conditions throughout the year, with routine traffic congestion being seasonal [20]. This would be especially expected out of Imperial County, as it has only experienced a population increase of 3.8% from 2010 to 2019 [8]. The lack of significant population growth, paired with rural road conditions such as lower average saturation flows, indicates that traffic conditions may be constant across days and times with minimal variability.

## The *Rational Agent Access (RAAM)* model

Though using travel times are useful indicators of access, spatial access models are far more comprehensive and are multilevel measures of access. There are currently six spatial access models, and each model measures different aspects of spatial accessibility; the most common model, the Floating Catchment Areas (FCA) model calculates the ratio of health providers to clients to a provider, within a given travel time to a hospital [15]. The oldest model, the Access Score (AS) model, finds the weighted sum of access components, such as the distance to a

hospital [21]. These models, though commonly used in spatial accessibility studies, lack complexity, and are limited in scope. Models such as FCA, for example, do not incorporate the distance from where a patient is traveling from in addition to the availability of resources and that there are multiple hospitals an individual can seek out.

To address the limitations of earlier accessibility models, this study used the Rational Agent Access Model (RAAM). The RAAM model is distinctly different from other spatial access models, as it takes consumer behavior and consumer demand for services in account when assessing access [22]. Eq (1) presents the mathematical formula used by the Rational Agent Access Model (RAAM),

$$\frac{\sum r'dr'l/sl}{\rho} + \frac{trl}{\tau} \tag{1}$$

where $s$ represents the supply of hospitals, $d$ represents the demand of hospital care, $l$ represents location of hospitals, $r$ represents the residential location of an agent, and $trl$ represents the travel cost or time taken from an agent's residence to the hospital. Regarding the other aspects of the equation, $dr'l$ represents optimized locations in which individuals actually seek hospital care, $\tau$ represents a normalized parameter for travel costs or time, and $\rho$ represents the national inverse primary care provider-to-patient ratio (PPR) which has a fixed value of 1,315 as of 2010 [22]. Travel times generated by RAAM account for daily traffic conditions and fluctuations, as they are based on average driving travel times in a region.

As per rational choice theory, individuals are economically rational beings who value efficiency and take information, costs and benefits and preferences to perform optional and efficient outcomes [23]. This model deploys this theory within a spatial context, assessing all possible outcomes and locations that a "rational agent" was to seek out based on spatial data from where demand for a service stem from. RAAM seeks to measure the saturation of people attempting to use a service, which is referred to as the "congestion" to a service. RAAM computes a spatial access score for its originating point by using the level of congestion and computing that in addition to the travel time from the point of origin to a particular service. In the formula, congestion is the supply and demand ratio, and it is scaled by the point of origin's average for the supply and demand of a service, which is essentially the normalized average for an area [22].

RAAM also incorporates an algorithm which seeks to find the lowest cost of congestion and travel time, ensuring that the score is realistically generated and considers willingness to travel for a particular service. In essence, RAAM assesses the allocation of a service (e.g., hospitals) and the population of an area using travel times and hospital congestion [22]. Travel times in the RAAM model are referred to as the "travel cost," which also ties into the ultimate score generated by the RAAM. Scores generated by RAAM represent the "cost" to access a particular service from a particular area; unlike other spatial access models such as FCA, RAAM represents cost rather than availability [22]. Higher scores generated by RAAM means that access is low, as there is a higher cost to the individuals in a region to access a service. To generate RAAM scores, this study used the PySAL Spatial Access Package LiveApp [24]. This is an open-source tool with a web-based interface that requires the number of emergency healthcare facilities per census tract, based on the U.S. Census 2010 shapefile.

## Results and discussion

Fig 1 depicts the travel times from each census tract centroid to Pioneers Memorial Hospital in Brawley, and Fig 2 depicts the travel times from each census tract centroid to El Centro Regional Medical Center in El Centro.

### Travel Times to Pioneers Memorial Hospital, Brawley

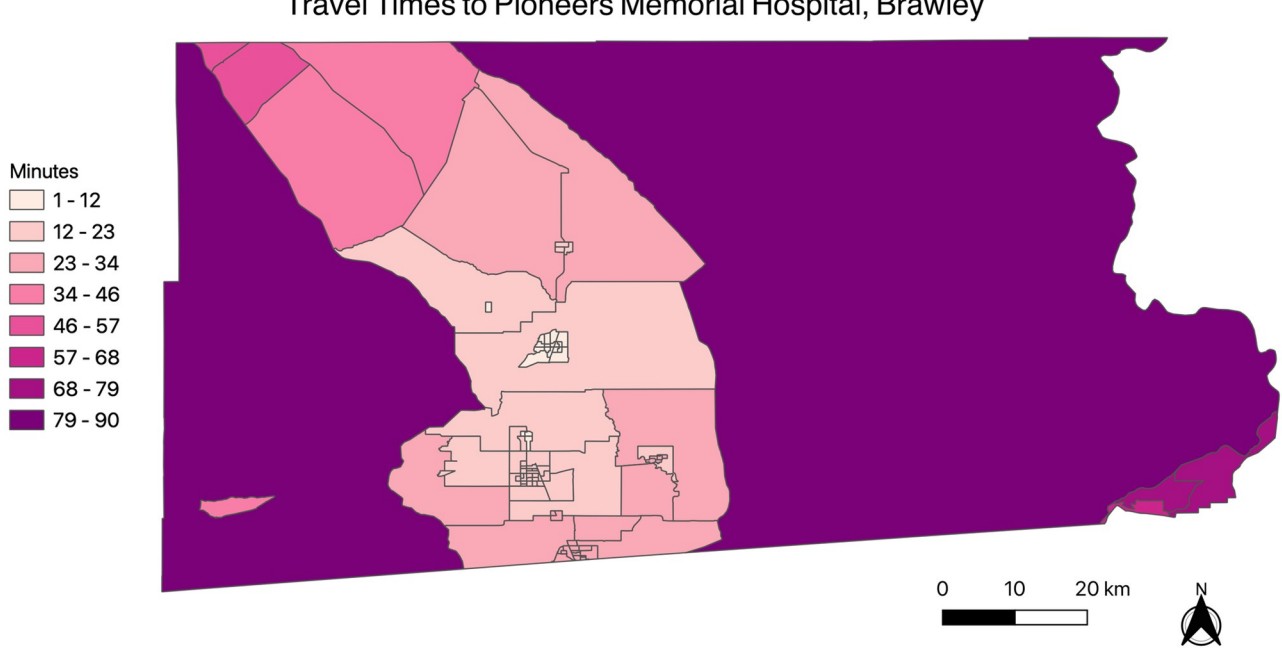

**Fig 1. Travel times between Imperial County census tract centroids and Pioneers Memorial Hospital, Brawley.** The map was generated using the free and open-source software QGIS version 3.14 (http://www.qgis.org/en/site/about/index.html).

### Travel Times to El Centro Regional Medical Center, El Centro

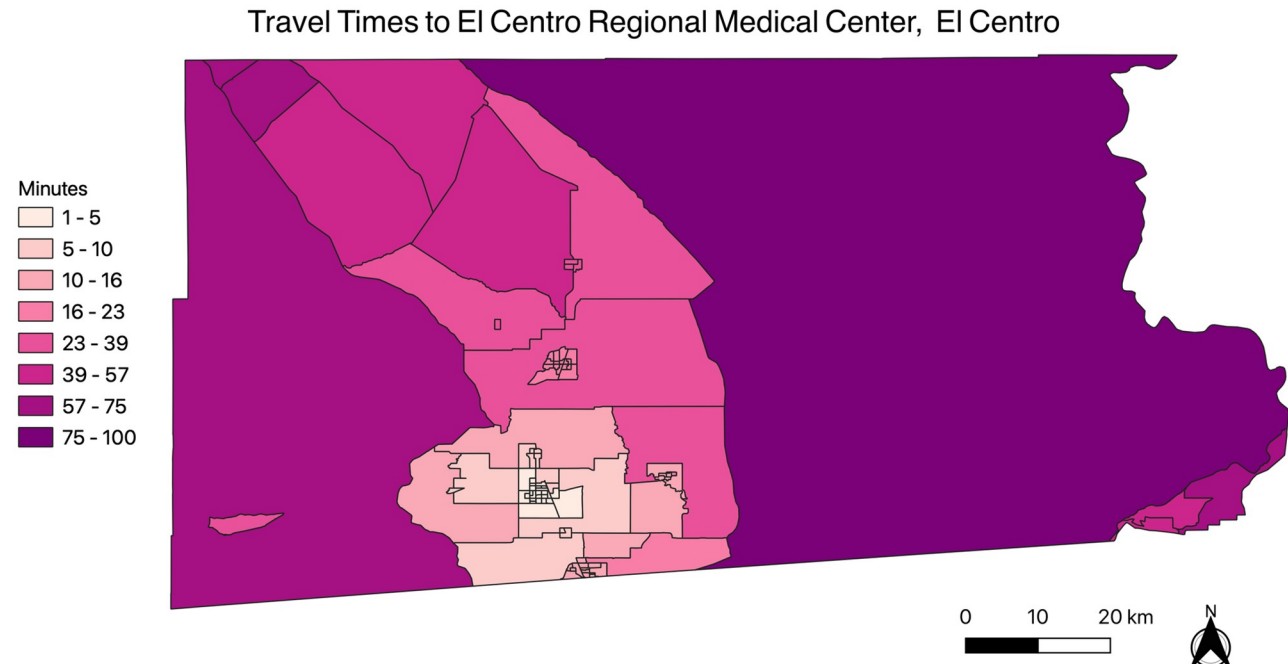

**Fig 2. Travel times between census tract centroids and El Centro Regional Medical Center, El Centro.** The map was generated using the free and open-source software QGIS version 3.14 (http://www.qgis.org/en/site/about/index.html).

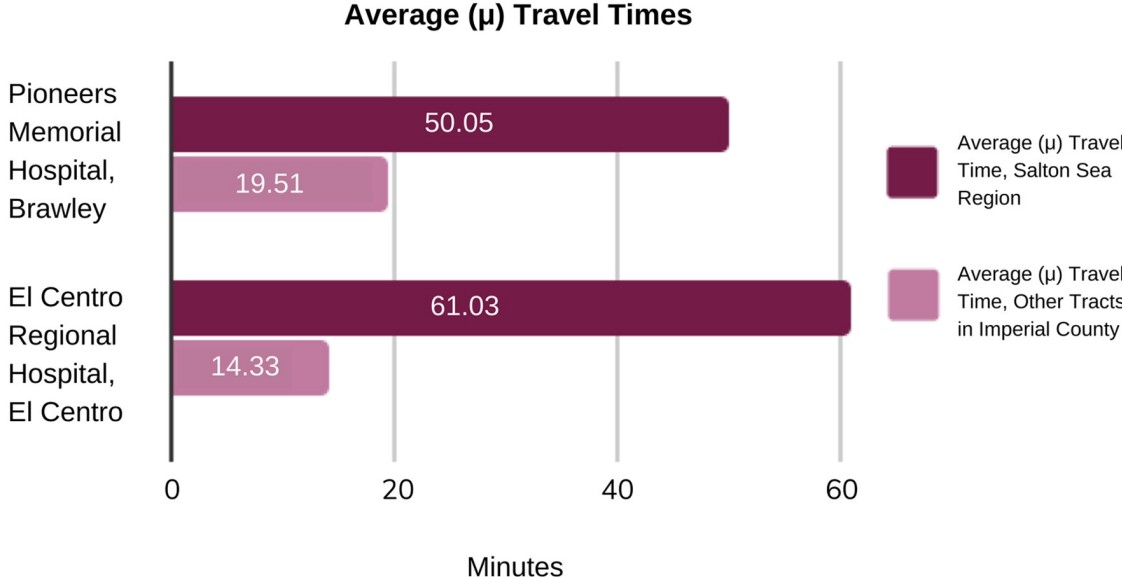

**Fig 3. Average (μ) travel times from census tracts to Imperial County Hospitals.**

The figures highlight that census tracts on the north, east and western regions of the county have higher travel times compared to census tracts in the south-central region of the county, which is where both county hospitals are both located. As seen in Fig 3, the average travel time for the Salton Sea region to Pioneers Memorial Hospital in Brawley is approximately 50 minutes, which is approximately 20 minutes higher, on average, than other census tracts in Imperial County. Similarly, the average travel time for the Salton Sea region to El Centro Regional Medical Center in El Centro is approximately 61 minutes, approximately 14 minutes higher, on average, than other Imperial County census tracts. This indicates that the Salton Sea region, on average, has higher travel times compared to other census tracts within Imperial County.

RAAM scores also reinforce this notion; Fig 4 depicts the score distribution per shapefile census tract in Imperial County. The average RAAM score for the Salton Sea region was approximately 1.80, whereas the average RAAM score for other Imperial County census tracts was approximately 1.33.

Since scores in the Salton Sea region were higher than other census tracts in the county, there is a higher cost for individuals in the area to reach and use emergency healthcare services. This means that for Salton Sea residents, there are limitations in travel costs in addition to an adequate number of emergency healthcare facilities to support population health needs. Since the supply of hospitals is low and accessible to individuals residing closer to the hospitals, there is excess competition and high demand between individuals in the region for emergency healthcare services. Therefore, spatial access to emergency healthcare services for Salton Sea residents of Imperial County is low. Though these scores are relative within the scope of this study, they lead to the conclusion that emergency healthcare access for the Salton Sea region is much more limited compared to other Imperial County census tracts.

Overall, assessment of spatial access to emergency healthcare facilities and services in Imperial County determined that access is low. This is due to high travel times and a low supply of hospitals that have the capacity to support residents in the area. Results from this study indicate that there is an increased need for emergency healthcare services in the Salton Sea Region of Imperial County. RAAM values are particularly indicative of this, as there is competing

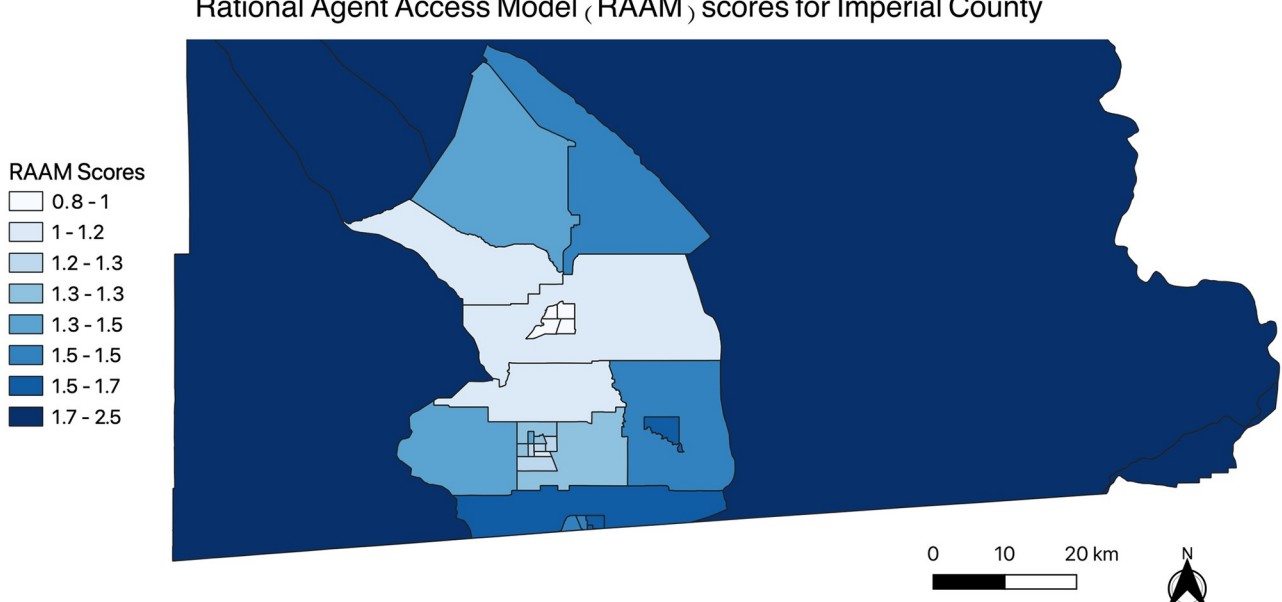

**Fig 4. Map of Rational Agent Access Model (RAAM) scores for Imperial County 2010 shapefile census tracts.** The map was generated using the free and open-source software QGIS version 3.14 (http://www.qgis.org/en/site/about/index.html).

demand for the few emergency healthcare services in the Salton Sea region. This study is distinct in that it is a practical and empirical application of rational choice theory within the context of spatial accessibility, particularly in an underserved and rural region. Pairing economic theory with traditional spatial accessibility measures such as travel times and hospital supply provides a more comprehensive and less static measure of resource accessibility.

## Policy recommendations and challenges

Based on the generated travel time and RAAM data, policies should be implemented to increase access to emergency healthcare in the Salton Sea region. However, the solution may not be as intuitive as building another hospital. Alternative forms of emergency healthcare services should be considered, as such services may be economically feasible and consider equity factors. Since existing literature does not discuss policy recommendations to address emergency healthcare access, the following policy recommendations are based on rural healthcare accessibility. Telehealth-enabled emergency healthcare services, for example, have proven to be helpful alternatives to traditional healthcare settings [25]. Existing literature suggests that these services help reduce ambulance transport to emergency health services, and telehealth is widely used in rural areas for providing healthcare [25]. However, the implementation of such a service is dependent on the triage score or acuity level of an emergency healthcare visit. Acuity is the level of intensity of a condition, and emergency rooms use triage scores to determine how serious the patient's complication is to address high acuity situations first [26]. A major challenge in finding appropriate policy recommendations is that current data surrounding the rate of emergency healthcare visits for pediatric asthma-related incidents do not break down the acuity level of the emergency care visits. Policymakers and regional hospitals should collect data on the acuity level of pediatric asthma-related emergency healthcare visits, as that would provide a much more accurate way of determining the types of emergency services needed in the Salton Sea region and Imperial County overall. If acuity cases are primarily low, using

emergency telehealth services may remove the need for Salton Sea residents to drive to an emergency room.

Another challenge with formulating policy to address emergency health facility access is that there is no information as to what days or times there is an influx of pediatric asthma-related emergency healthcare visits. There is also no existing work that assesses emergency visit rates and their relation to the levels of particulate matter in the air. Information regarding where pediatric asthma rates is higher is unavailable, which poses a challenge in determining which regions need higher healthcare access. Additionally, information regarding where individuals who require emergency healthcare services due to pediatric asthma-related incidents are originally coming from is unavailable. Though the Salton Sea region is more vulnerable than other areas in Imperial County, there is a possibility that the asthma rates and pediatric asthma-related emergency healthcare visits are clustered in other regions. Creating a system that tracks pediatric asthma hotspots and how air quality fluctuations influence asthma-related emergency healthcare visits would assist in locating regions that require more healthcare support. Services such as EMT stations could help stabilize patients in such regions by providing them with instant care in their original region. Ambulance service providers in England treat and discharge patients after attending to them within the ambulance itself, reducing the number of emergency healthcare facility visits [27]. This type of service may allow for increased healthcare access as well as a reduced time in traveling to a hospital in rural regions like Imperial County.

It is important to note, however, that RAAM values and generated travel times cannot be solely used to assess resource accessibility and influence policy to increase emergency healthcare access. Though the RAAM model poses many benefits and is more comprehensive than previous accessibility models, it does not account for transportation access or social demand. The RAAM model assists in understanding which regions require a higher quantity of services than in the present, but the equitable allocation of resources is ultimately dependent on socioeconomic (SES) factors. Within the Salton Sea region, there may be census tracts that need additional support due to higher rates of poverty, age-related factors, and a lack of infrastructure. Social demand assesses the needs of disadvantaged groups such as low-income, elderly, and disabled groups and newer work has also taken socioeconomic output and population density into account [28]. Assessing social demand when pursuing policies to bolster healthcare access may be helpful in determining which areas within the Salton Sea region itself have a higher need for services.

Social demand also plays a role in understanding the relationship between service access and transportation access, as transportation access influences an individual's ability to reach services such as hospitals. It is therefore equally important to understand how accessible public transportation is, and if resources are being equitably distributed to regions with a higher need for transportation. There is existing work that has assessed the relationship between social demand and transportation accessibility and has found that such assessments are vital in urban planning and transportation infrastructural development [28]. Though this is beyond the scope of this paper, future work should consider the relationship between transportation access and social demand, as such an analysis could directly influence policy regarding transportation infrastructure in the region to increase access to hospitals.

## Conclusion

Imperial County's Salton Sea region had low levels of emergency healthcare accessibility due to high travel times as well as competing demand for the emergency healthcare services, relative to population and service congestion. Data limitations and policy challenges may impact

regional policymakers to address the issue of emergency healthcare service access, unless addressed soon. Though interventions to address and tackle emergency healthcare facility access are essential to ensure that pediatric asthma patients have efficient access to health facilities, they would not address pollution and particulate matter exacerbating asthma symptoms. They would also not address income inequities, which may deter or prohibit individuals experiencing severe asthma complications from acquiring medical treatment. Ultimately, it is the intervention of the state and regional municipalities to address the environmental health crisis in the area that may lead to a decrease in emergency healthcare related visits associated with asthma in Imperial County and the Salton Sea region.

## Acknowledgments

I would like to thank the anonymous reviewers for their careful reviews and comments. I would also like to thank the editors' help provided during the editorial process. Lastly, I would like to thank the continued mentorship and support by the faculty and staff at both the University of California, Riverside Center for Geospatial Sciences (UCR CGS) and the University of California, Riverside Center for Health Disparities Research (UCR HDR@UCR).

## Author Contributions

**Conceptualization:** Preeti Juturu.

**Data curation:** Preeti Juturu.

**Formal analysis:** Preeti Juturu.

**Funding acquisition:** Preeti Juturu.

**Investigation:** Preeti Juturu.

**Methodology:** Preeti Juturu.

**Project administration:** Preeti Juturu.

**Resources:** Preeti Juturu.

**Software:** Preeti Juturu.

**Supervision:** Preeti Juturu.

**Validation:** Preeti Juturu.

**Visualization:** Preeti Juturu.

**Writing – original draft:** Preeti Juturu.

**Writing – review & editing:** Preeti Juturu.

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
