## [Decision Letter · Decision Letter 0]

8 Apr 2021

PONE-D-21-07464

Assessing Emergency Healthcare Accessibility in the Salton Sea Region of Imperial County, California

PLOS ONE

Dear Dr. Juturu,

Thank you for submitting your manuscript to PLOS ONE. After careful consideration, we feel that it has merit but does not fully meet PLOS ONE’s publication criteria as it currently stands. Therefore, we invite you to submit a revised version of the manuscript that addresses the points raised during the review process.

We look forward to receiving your revised manuscript.

Kind regards,

Jun Yang

Academic Editor

PLOS ONE

Journal Requirements:

2. We note that Figures 1, 2, 3 and 5 in your submission contain map images which may be copyrighted.

a. You may seek permission from the original copyright holder of Figures 1, 2, 3 and 5 to publish the content specifically under the CC BY 4.0 license. 

Additional Editor Comments:

Reviewer 1

The paper used RAAM to assess the level of emergency healthcare access in the Salton Sea region of Imperial County. In addition to travel times, the supply and demand for hospitals were taken into account. The method is novel and the study has practical application value. But there are still some problems:

1. Introduction. It should include reviews of other studies assessing the level of emergency healthcare access and explain the gaps to reflect the research value and innovation of this paper.

2. In Fig.5, it is suggested to increase the number of grades of the RAAM Scores to make the visualization effect better. The current results seem to be relatively general and cannot correspond well with the results in Fig.3. Also, the clarity of all figures should be improved.

3. The paper is of practical value, and the author gives policy suggestion. However, the content and analysis are not rich enough to meet the requirements of a research article. It is recommended to add deeper analysis and interpretation corresponding with the Policy Challenges section. In addition, the innovation of the study needs to be explained: whether the data/indicators are more perfect, or the method is better, etc. Most importantly, what are the supplements to the previous studies in this field. Please elaborate the innovation in the Introduction or Conclusion section.

Reviewer 2

The authors assessed the emergency healthcare accessibility in the Salton Sea region of Imperial County. The research methodologies are reasonable, and the findings are interesting. However, there are still a few aspects that should be improved to make the paper publishable. As the article is coherent and well-organized, I focus here only on some points, which are hopefully easy for the authors to take into account in the revision.

1. North arrow should be added to figures.

2. This work used 3:00 to 4:00 PM as a reference time to assess the accessibility. What about accessibility in the normal time? The comparison should be discussed because emergencies are random.

3. There are also some methods to evaluate accessibility pattern, and the advantages and disadvantages need to be emphasized. Suggest reading, for instance,

(1) The influence of high-speed rail on ice–snow tourism in northeastern China. Tourism Management,2020, doi:10.1016/j.tourman.2019.104070.

(2) Study on the Impact of High-speed Railway Opening on China's Accessibility Pattern and Spatial Equality[J]. Sustainability, 2018,10,2943. doi:10.3390/su10082943

Reviewers' comments:

Reviewer's Responses to Questions

**Comments to the Author**

1. Is the manuscript technically sound, and do the data support the conclusions?

Reviewer #1: Yes

Reviewer #2: Yes

2. Has the statistical analysis been performed appropriately and rigorously? 

Reviewer #1: Yes

Reviewer #2: Yes

3. Have the authors made all data underlying the findings in their manuscript fully available?

Reviewer #1: Yes

Reviewer #2: Yes

4. Is the manuscript presented in an intelligible fashion and written in standard English?

Reviewer #1: Yes

Reviewer #2: Yes

5. Review Comments to the Author

Reviewer #1: The paper used RAAM to assess the level of emergency healthcare access in the Salton Sea region of Imperial County. In addition to travel times, the supply and demand for hospitals were taken into account. The method is novel and the study has practical application value. But there are still some problems:

1. Introduction. It should include reviews of other studies assessing the level of emergency healthcare access and explain the gaps to reflect the research value and innovation of this paper.

2. In Fig.5, it is suggested to increase the number of grades of the RAAM Scores to make the visualization effect better. The current results seem to be relatively general and cannot correspond well with the results in Fig.3. Also, the clarity of all figures should be improved.

3. The paper is of practical value, and the author gives policy suggestion. However, the content and analysis are not rich enough to meet the requirements of a research article. It is recommended to add deeper analysis and interpretation corresponding with the Policy Challenges section. In addition, the innovation of the study needs to be explained: whether the data/indicators are more perfect, or the method is better, etc. Most importantly, what are the supplements to the previous studies in this field. Please elaborate the innovation in the Introduction or Conclusion section.

Reviewer #2: The authors assessed the emergency healthcare accessibility in the Salton Sea region of Imperial County. The research methodologies are reasonable, and the findings are interesting. However, there are still a few aspects that should be improved to make the paper publishable. As the article is coherent and well-organized, I focus here only on some points, which are hopefully easy for the authors to take into account in the revision.

1.North arrow should be added to figures.

2.This work used 3:00 to 4:00 PM as a reference time to assess the accessibility. What about accessibility in the normal time? The comparison should be discussed because emergencies are random.

3.There are also some methods to evaluate accessibility pattern, and the advantages and disadvantages need to be emphasized. Suggest reading, for instance,

(1)The influence of high-speed rail on ice–snow tourism in northeastern China. Tourism Management,2020, doi:10.1016/j.tourman.2019.104070.

(2)Study on the Impact of High-speed Railway Opening on China's Accessibility Pattern and Spatial Equality[J]. Sustainability, 2018,10,2943. doi:10.3390/su10082943

6. PLOS authors have the option to publish the peer review history of their article (what does this mean?). If published, this will include your full peer review and any attached files.

Reviewer #1: No

Reviewer #2: No

---

## [Author Response · Author response to Decision Letter 0]

28 May 2021

Journal Requirements:

Comment-1. Please ensure that your manuscript meets PLOS ONE's style requirements, including those for file naming. The PLOS ONE style templates can be found at https://journals.plos.org/plosone/s/file?id=wjVg/PLOSOne_formatting_sample_main_body.pdf and https://journals.plos.org/plosone/s/file?id=ba62/PLOSOne_formatting_sample_title_authors_affiliations.pdf

Response: Thank you for your feedback. This manuscript has been revised according to the journal style requirements (please see the revised manuscript).

Comment-2. We note that Figures 1, 2, 3 and 5 in your submission contain map images which may be copyrighted. All PLOS content is published under the Creative Commons Attribution License (CC BY 4.0), which means that the manuscript, images, and Supporting Information files will be freely available online, and any third party is permitted to access, download, copy, distribute, and use these materials in any way, even commercially, with proper attribution. For these reasons, we cannot publish previously copyrighted maps or satellite images created using proprietary data, such as Google software (Google Maps, Street View, and Earth). For more information, see our copyright guidelines: http://journals.plos.org/plosone/s/licenses-and-copyright.

a. You may seek permission from the original copyright holder of Figures 1, 2, 3 and 5 to publish the content specifically under the CC BY 4.0 license. 

USGS National Map Viewer (public domain): 

http://viewer.nationalmap.gov/viewer/

The Gateway to Astronaut Photography of Earth (public domain): 

http://eol.jsc.nasa.gov/sseop/clickmap/

Maps at the CIA (public domain): 

https://www.cia.gov/library/publications/the-world-factbook/index.html and 

https://www.cia.gov/library/publications/cia-maps-publications/index.html

Natural Earth (public domain): 

http://www.naturalearthdata.com/

Response: Thank you for bringing this to my attention. The figures have been revised using GQIS, so figures are copyright-free and are formatted as per journal requirements (please see the revised figures).

Comment-3. Please amend either the title on the online submission form (via Edit Submission) or the title in the manuscript so that they are identical.

Response: Thank you for your feedback. I have incorporated your comments by renaming the manuscript title (please see the revised manuscript).

Response to Reviewer #1 comments:

The paper used RAAM to assess the level of emergency healthcare access in the Salton Sea region of Imperial County. In addition to travel times, the supply and demand for hospitals were taken into account. The method is novel and the study has practical application value. But there are still some problems: 

Comment-1: Introduction. It should include reviews of other studies assessing the level of emergency healthcare access and explain the gaps to reflect the research value and innovation of this paper. 

Response: Thank you for your valuable insight. I’ve attempted to address these items by discussing existing literature on rural emergency healthcare access. I’ve also included current gaps in existing research and ways in which my work supplements and expands upon current literature. However, I was unable to include many reviews as most work does not focus on rural regions. Therefore, this section in the introduction may be a bit shorter than other works. 

Comment-2: In Fig.5, it is suggested to increase the number of grades of the RAAM Scores to make the visualization effect better. The current results seem to be relatively general and cannot correspond well with the results in Fig.3. Also, the clarity of all figures should be improved.

Response: Thank you for your suggestions. I agree with you and have incorporated this suggestion by re-visualizing my data for improved figure clarity. I have also increased the number of grades of the RAAM values, so they are clearer to visualize and interpret (please see the revised figures).

Comment-3: The paper is of practical value, and the author gives policy suggestion. However, the content and analysis are not rich enough to meet the requirements of a research article. It is recommended to add deeper analysis and interpretation corresponding with the Policy Challenges section. In addition, the innovation of the study needs to be explained: whether the data/indicators are more perfect, or the method is better, etc. Most importantly, what are the supplements to the previous studies in this field. Please elaborate the innovation in the Introduction or Conclusion section.

Response: Thank you for providing these insights. I have incorporated your comment about policy analysis and interpretation in my work. This has been done by drawing from existing research that assesses emergency healthcare access in rural areas and gauging the effectiveness of the previous healthcare access policies. However, there are certain limitations in doing so as academic literature and research on policy recommendations to address spatial access to rural emergency healthcare facilities are limited in nature. This study arguably seeks to bring attention to the spatial inaccessibility to healthcare facilities in the region at hand and brings awareness to potential solutions that may be deployed to address spatial healthcare accessibility. Therefore, I speak primarily about policy examples that may be helpful in addressing emergency healthcare access. Regarding the study innovation, I attempt to discuss the ways in which my work expands upon current work, and I incorporate this within my materials and methods section in addition to the results and discussion section.

Response to Reviewer #2 comments:

The authors assessed the emergency healthcare accessibility in the Salton Sea region of Imperial County. The research methodologies are reasonable, and the findings are interesting. However, there are still a few aspects that should be improved to make the paper publishable. As the article is coherent and well-organized, I focus here only on some points, which are hopefully easy for the authors to take into account in the revision. 

Comment-1: North arrow should be added to figures.

Response: Thank you for your suggestion. I have incorporated this feedback into my newly produced figures (please see the revised figures).

Comment-2: This work used 3:00 to 4:00 PM as a reference time to assess the accessibility. What about accessibility in the normal time? The comparison should be discussed because emergencies are random.

Response: Thank you for raising this important question. In this work, I use the 3:00 to 4:00 PM reference time as a case study to generate and analyze travel times. Though this is a static time frame, which may arguably be limiting in nature considering the unpredictability of emergency health situations, rural regions such as Imperial County experience constant traffic conditions throughout the year, with routine traffic congestion being seasonal (Gastelle et al., 2017). This would be especially expected out of Imperial County, as it has only experienced a population increase of 3.8% from 2010 to 2019 (U.S. Census Bureau, 2019). The lack of significant population growth, paired with rural road conditions such as lower average saturation flows, indicates that traffic conditions may be constant across days and times with minimal variability. However, to ensure results are generalizable, the RAAM model uses average driving travel times to calculate RAAM values. Therefore, RAAM values are representative of annual and daily fluctuations and can be used to compare case-study results. Interestingly, the travel times choropleth map is closely aligned with the RAAM values map, which may suggest that traffic conditions in the region remain constant. Though that is not necessarily the scope of the study, this correlation has emerged and may be helpful in studying further to determine appropriate policy solutions. I have added this information and its corresponding academic sources to my manuscript to address your question, which is deeply appreciated.

Comment-3: There are also some methods to evaluate accessibility pattern, and the advantages and disadvantages need to be emphasized. Suggest reading, for instance, (1) The influence of high-speed rail on ice–snow tourism in northeastern China. Tourism Management,2020, doi:10.1016/j.tourman.2019.104070. (2) Study on the Impact of High-speed Railway Opening on China's Accessibility Pattern and Spatial Equality[J]. Sustainability, 2018,10,2943. doi:10.3390/su10082943 

Response: Thank you for your feedback and for providing me with these resources. I have had the opportunity to review and draw information from them regarding social demand; this information has been incorporated into the results and discussion section. However, incorporating materials on accessibility patterns and the various methods in-depth may be beyond the scope of this paper. The focus for this work is to assess spatial emergency healthcare access within Imperial County with a particular focus on the Salton Sea region. The emphasis for this study is on the RAAM model and economic demand, as opposed to personal resources or demographics. However, I will be sure to consider it for future work, such as conducting a multivariate analysis of emergency healthcare access patterns and spatial equity.

---

## [Decision Letter · Decision Letter 1]

2 Jun 2021

Assessing emergency healthcare accessibility in the Salton Sea region of Imperial County, California

PONE-D-21-07464R1

Dear Dr. Juturu,

We’re pleased to inform you that your manuscript has been judged scientifically suitable for publication and will be formally accepted for publication once it meets all outstanding technical requirements.

Kind regards,

Jun Yang

Academic Editor

PLOS ONE

Additional Editor Comments (optional):

Accept

Reviewers' comments:

Reviewer's Responses to Questions

**Comments to the Author**

1. If the authors have adequately addressed your comments raised in a previous round of review and you feel that this manuscript is now acceptable for publication, you may indicate that here to bypass the “Comments to the Author” section, enter your conflict of interest statement in the “Confidential to Editor” section, and submit your "Accept" recommendation.

Reviewer #1: All comments have been addressed

Reviewer #2: All comments have been addressed

2. Is the manuscript technically sound, and do the data support the conclusions?

Reviewer #1: Yes

Reviewer #2: Yes

3. Has the statistical analysis been performed appropriately and rigorously? 

Reviewer #1: Yes

Reviewer #2: Yes

4. Have the authors made all data underlying the findings in their manuscript fully available?

Reviewer #1: Yes

Reviewer #2: Yes

5. Is the manuscript presented in an intelligible fashion and written in standard English?

Reviewer #1: Yes

Reviewer #2: Yes

6. Review Comments to the Author

Reviewer #1: (No Response)

Reviewer #2: Authors have taken into account my suggestions and comments. I believe the current version can be published.

7. PLOS authors have the option to publish the peer review history of their article (what does this mean?). If published, this will include your full peer review and any attached files.

Reviewer #1: No

Reviewer #2: No

---

## [Editor Report · Acceptance letter]

7 Jun 2021

PONE-D-21-07464R1 

Assessing emergency healthcare accessibility in the Salton Sea region of Imperial County, California 

Dear Dr. Juturu:

I'm pleased to inform you that your manuscript has been deemed suitable for publication in PLOS ONE. Congratulations! Your manuscript is now with our production department. 

Kind regards, 

on behalf of

Dr. Jun Yang 

Academic Editor

PLOS ONE